# MgO Heterostructures: From Synthesis to Applications

**DOI:** 10.3390/nano12152668

**Published:** 2022-08-03

**Authors:** Tabasum Huma, Nadimullah Hakimi, Muhammad Younis, Tanzeel Huma, Zhenhua Ge, Jing Feng

**Affiliations:** 1Faculty of Material Science and Engineering, Kunming University of Science and Technology, Kunming 650093, China; tabassumhumaanwar@gmail.com (T.H.); nadim.hakimi123@gmail.com (N.H.); zge@kust.edu.cn (Z.G.); 2Department of Polymeric Materials, School of Materials Science and Engineering, Beijing Institute of Technology, No. 5, Zhongguancun South Street, Beijing 100081, China; youniskhant20@yahoo.com; 3Yale School of Medicine, Yale University, New Haven, CT 06520, USA; tanzeelhuma@outlook.com

**Keywords:** MgO, heterostructures, magnetic storage, energy storage

## Abstract

The energy storage capacity of batteries and supercapacitors has seen rising demand and problems as large-scale energy storage systems and electric gadgets have become more widely adopted. With the development of nano-scale materials, the electrodes of these devices have changed dramatically. Heterostructure materials have gained increased interest as next-generation materials due to their unique interfaces, resilient structures and synergistic effects, providing the capacity to improve energy/power outputs and battery longevity. This review focuses on the role of MgO in heterostructured magnetic and energy storage devices and their applications and synthetic strategies. The role of metal oxides in manufacturing heterostructures has received much attention, especially MgO. Heterostructures have stronger interactions between tightly packed interfaces and perform better than single structures. Due to their typical physical and chemical properties, MgO heterostructures have made a breakthrough in energy storage. In perpendicularly magnetized heterostructures, the MgO’s thickness significantly affects the magnetic properties, which is good news for the next generation of high-speed magnetic storage devices.

## 1. Introduction

The overconsumption of fossil fuels drives us to create and use more renewable energy in our daily lives. Various electrochemical energy storage systems, such as lithium-ion batteries, sodium-ion batteries and potassium-ion batteries (KIBs), have been created in the last two decades to store and utilize this energy at any time. The performance of these devices mainly depends on the materials used to make the electrodes. Graphite, hard carbon, metal oxides, sulfides and carbides, among other unique materials, have shown remarkable electrochemical performance when used as electrodes in energy storage devices [1], as shown in (Figure 1).

Like a coin with two faces, each material has its drawbacks that limit its ability to perform at its peak. The scientific community has studied high-performance electrode materials’ properties and design principles, including structural design, growth on a free-standing substrate and protective layer coating, which considerably enhances the energy storage capacity of new materials. Creating high-performance electrodes by building heterostructures is a new concept that has just been introduced and is now undergoing successful development. Most heterostructure electrodes outperform their parts in reversible capacity and cycle stability and some even surpass the theoretical capacity limit, proving the superiority of this concept. Looking back at the history of condensed matter physics, we see that the notion of the heterostructure was not originally presented in the context of energy storage. W. Shockley initially offered this idea for wide-gap semiconductor emitter applications in semiconductor physics [2,3,4]. Different semiconductors with identical crystal structures, atomic spacing and thermal expansion coefficients make up the heterostructure. A heterostructure’s chemical composition and charge distribution can alter across this interface due to the multiple band structures, predominant carrier concentration differential and Fermi-level differences in the band structure [5].

Heterostructures are semiconductor structures that have a one-of-a-kind natural organization. A heterostructure is an essential structure made up of a mixture of two different semiconductor materials as shown in (Figure 2). The materials have inconsistent bandgaps [6,7].

It has three types: the type-I heterostructure (straddling gap), the type-II heterostructure (staggered gap) and the type-III heterostructure (broken gap) [8,9]. The staggering band structure of type-II heterostructures causes the photo-excited carriers to be effectively separated into different monolayers, lowering the recombination rate [10,11]. Heterostructures can be made of several materials with a tight interface, and the properties of these structures can be changed by altering their physicochemical properties. Heterostructures are classified based on the material configuration and interface, as follows: (1) spheric, (2) cylindrical, (3) planar and (4) cubic. Because nanomaterials exhibit unusual properties at the 0D level, they have become a promising area of research. In nanoscale structures, such as nanoclusters and nano dispersions, central shelled structures, quantum wells, and quantum nodes are the most common 0D heterostructures [12,13].

**Figure 2 nanomaterials-12-02668-f002:**
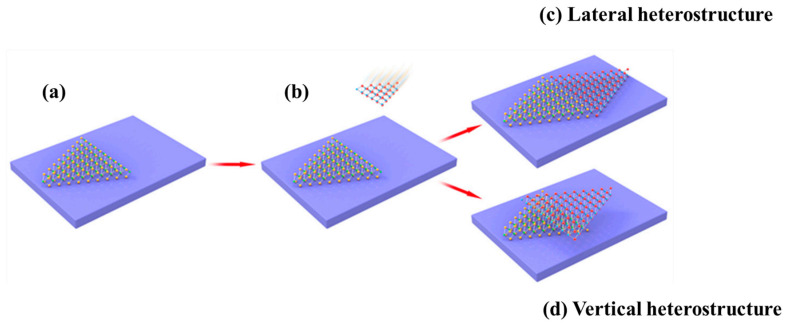
The preparation of lateral and vertical heterostructures is depicted in a schematic diagram. (**a**) The first TMD layer on the base. (**b**) The construction of a second TMD layer using a first TMD layer. (**c**) Two different forms of TMDs build lateral heterostructure. (**d**) Vertical heterostructure is constructed using two kinds of TMDs [14].

## 2. Metal Oxide

Metal oxides are a class of materials that are formed when the elements of metal and oxygen interact. Some nanomaterials have been found to act as solid catalysts, and their catalytic activity increases when they are reduced to nanoscale sizes. When empty voids or pores are introduced into metal oxide structures, the materials become even more beneficial for use in various applications. Their applications allow them to be combined with other materials, enabling them to take advantage of each other’s properties. Metal oxide porosity can be classified as microporous, mesoporous, or macroporous if the pore sizes are 2 nm, 2–50 nm or greater than 50 nm, respectively [15,16]. The band gap energies of different metal oxides are MoO_2_ (1.74 eV), MoO_3_ (2.94 eV), ZnO (3.2 eV), CuO (1.7 eV), TiO_2_ (3.2 eV) and Fe_2_O_3_ (2.14 eV). Their optical and electronic properties make them useful as 2D materials in photodetectors as well as in high-temperature electronic devices. They are used for a variety of applications, including the oxidation of toxic pollutants as well as the photocatalytic degradation of dyes in waste water. Their increased surface area and reactive sites create sufficient heat energy and chemical agents to degrade pollutants [17]. The use of metal oxide nanostructures, such as cerium oxide, zinc oxide, iron oxide, tin oxide, zirconium oxide, titanium oxide, and magnesium oxide in biocompatible coatings and catalysts, has been shown to have potential applications in a wide range of industries [18].

Magnesium oxide (MgO) is a well-known insulator with a wide bandgap of 7.8 eV. It is appealing for use in insulation applications due to its low heat capacity and high melting point [19]. Magnesium oxide nanostructures have also been used as protective layers for dielectrics in AC circuits due to their anti-sputtering properties, high transmittance and secondary electron emission coefficient [20].

## 3. Magnesium Oxide

### 3.1. Basic

Magnesium oxide, commonly known as periclase [21], is a hygroscopic white solid mineral derived from the Greek term “periklao”, meaning “around” and “to cut”. Magnesium oxide has the empirical formula MgO and its lattice is made up of magnesium ions and oxygen ions linked by an ionic bond. Magnesium oxide is produced through the calcination of magnesium hydroxide Mg(OH)_2_ or magnesium carbonate MgCO_3_. The surface area and pore size of the formed magnesium oxide as well as the final reactivity are all affected by the thermal treatment used during the calcination process. The calcining of magnesium oxide occurs between 700 and 1000 degrees Celsius, forming caustic calcined magnesium oxide between 1000 and 1500 degrees where lower chemical activity magnesium oxide is formed, and above 1500 degrees where refractory magnesium oxide with reduced chemical activity is formed. This refractory magnesium oxide is used primarily for electrical and refractory applications [22].

Magnesium oxide has many uses; it is used in catalysts and to remediate toxic waste, and is added to refractory products, paint and superconductor materials. Magnesium oxide is an insulator; however, one-dimensional magnesium oxide nanostructures can exhibit violet blue and blue-green phosphorescence [23,24,25,26,27,28,29,30] and are used in miniaturized optical devices. The effects of dopants on the PL properties of one-dimensional MgO nanostructures have been reported [25,26].

### 3.2. MgO Properties

Magnesium oxide has physical qualities that make it a viable choice for a variety of applications [22]. It comes in a variety of colors including white, brown and black (depending on the presence of iron or another foreign element). When considering the outermost layers of magnesium oxide, it becomes clear that it has the simplest oxide structure known as the rock-salt structure. It has a density of 3.579 g per cubic centimeter and a Mohs hardness of 5 on the scale. At 100 °C, the thermal conductivity of sintered magnesium oxide is 36 W/m^2^ (mK). Magnesium oxide has extremely high melting and boiling points due to its refractory qualities (melting point 2800 °C and boiling point 3600 °C). The purity of magnesium oxide determines the value of electrical resistance. Electrical resistivity values for high purity magnesia can approach 1016 Ωm. Specific resistance is mostly determined by chemical purity, although at higher temperatures such as 2000 °C and above, the purity of magnesia has no bearing on electrical resistivity values. Magnesium oxide has a dielectric constant ranging from 3.2 to 9.8 at 25 °C and 1 MHz and dielectric loss values for the same conditions are approximately 10^−4^ [22].

Magnesium oxide has a variety of uses in many different industries. Fireproofing elements in construction materials are valued for their resistance to heat [27]. Corrosion is also not an option in sectors like nuclear, chemical and superalloys [28]. Magnesium oxide is used as an antacid, laxative and magnesium supplement. It is also used to treat heartburn and sour stomach, as well as for insulators [31], fertilizers [32], water treatment [33] and protective coatings [34]. There is currently a movement to utilize nanoscale fillers [35]. Nanotechnology is defined as creating functional structures in the range of 0.1–100 nm using various physical or chemical methods [36]. That is true for magnesium oxide as well. The sol–gel method [31] or the hydrothermal method [32] can be utilized to make nanoscale magnesium oxide. MgO is a potential electrical insulator for high-voltage applications such as insulation. The large band gap (7.8 eV) and high-volume resistivity (1017 W/m) contribute to its suitability. Nanoscale oxides have the highest volume resistivity value [33]. NanoAmor produces magnesium oxide powder with an average diameter of 20 nm, a specific surface area of more than 60 m^2^/g and a density of 0.3 g/cm^3^.

Magnesium oxide nanoparticles are non-toxic, relatively inexpensive, and easy to manufacture. They usually have a particle size of 5–100 nanometers and a surface area of 25–50 m^2^/g. Despite their tiny size, metal oxide nanoparticles have higher melting and boiling points than bulk metal oxides [34]. Magnesium oxide, a compound with a cubic structure, has an empirical formula of MgO and a lattice arrangement of Mg^2+^ ions and O^2−^ ions in ionic bonding [37]. While magnesium oxide nanoparticles are only slightly soluble in water at 20 degrees Celsius, they can be used to create products with properties such as dust repellence, wear-resistance and fire-resistance, along with high intensity, hardness and thermal insulation [38].

### 3.3. Fabrication Techniques

#### 3.3.1. Chemical Vapor Deposition

Chemical vapor deposition (CVD) synthesis is a bottom-up approach to preparing vertical 2D heterostructures that can also be used to grow planar multi-junction heterostructures. Various techniques have been developed for growing vertically stacked heterostructures with atomically sharp interfaces and clean surfaces, as well as for fabricating planar heterostructures with controllable properties [39]. The magnesium oxide nanowires were principally synthesized through a method called chemical vapor deposition (CVD) and pulsed laser deposition (PLD). The CVD method is used to produce complex oxides for a variety of applications. While physical vapor deposition techniques allow for more flexibility in composition changes and are compatible with future device fabrication technologies, they are limited by the control of the composition, which is a major challenge in CVD. In response to these concerns, a liquid injection delivery scheme has been proposed [40]. The technique can be used to grow oxides at the nanometer scale and has been applied to epitaxially grow a variety of complex functional oxides including ferroelectrics and multiferroics [41,42].

#### 3.3.2. Magnetron Sputtering

One of the most common methods is magnetron sputtering. Magnets are used to trap secondary electrons, keeping them close to the target and increasing the plasma ionization rate, allowing for faster deposition rates. Using a magnetron sputtering system, one can coat large substrates with thin films made of materials such as gold and aluminum that have excellent adhesion on the support surface and uniform coverage of steps and small features (Figure 3b) [43].

#### 3.3.3. Atomic Layer Deposition (ALD)

The atomic layer deposition process is used in conjunction with template-assisted procedures to build metal oxide heterostructures. The precursors are pulsed and chemisorbed on the patterned substrate during the ALD process, which takes place in a reactor. The precursor material can take the shape of a liquid, solid or gas. The structure and morphology of the produced materials are influenced by the composition and chemistry of the vapor phase. The deposition temperature influences the material adherence and growth [44]. Xue et al. reported the fabrication of MgO/InP heterostructures by means of atomic layer deposition. The findings suggest that MgO/InP heterojunctions could be useful for electrical devices based on InP substrates. The calculated band gap of MgO is large enough to reduce the leakage current across the gap (Figure 3c) [45].

#### 3.3.4. Hydrothermal/Solvothermal Synthesis

One of the most explored methods is hydrothermal synthesis, which allows for the creation of a wide range of metal oxide heterojunctions. The chemical reactions and solubility changes of compounds in aqueous solution that occur above ambient temperature and pressure are the basis for this process. The synthesis technique is carried out in an autoclave, which allows for the precise control of the solution content, reaction temperature and time [46]. The solvothermal approach has similarities with hydrothermal synthesis, except that the solvent used to prepare the precursors is not aqueous. Shiming et al. used a simple two-step solvothermal approach to successfully construct heterostructured TiO_2_/MgO NWAs with lengths of 32 m on FTO glass. The development of a novel self-powered UVPD based on the photoelectrochemical cell using heterostructured TiO_2_/MgO nanowires was reported. By combining the heterostructured nanowire arrays with an electrolyte, the device shows a greater open-circuit voltage and photocurrent density than devices made of pure TiO_2_ nanowires, which is attributed to less charge recombination at the TiO_2_/MgO nanowire/electrolyte interfaces (Figure 3d) [47].

**Figure 3 nanomaterials-12-02668-f003:**
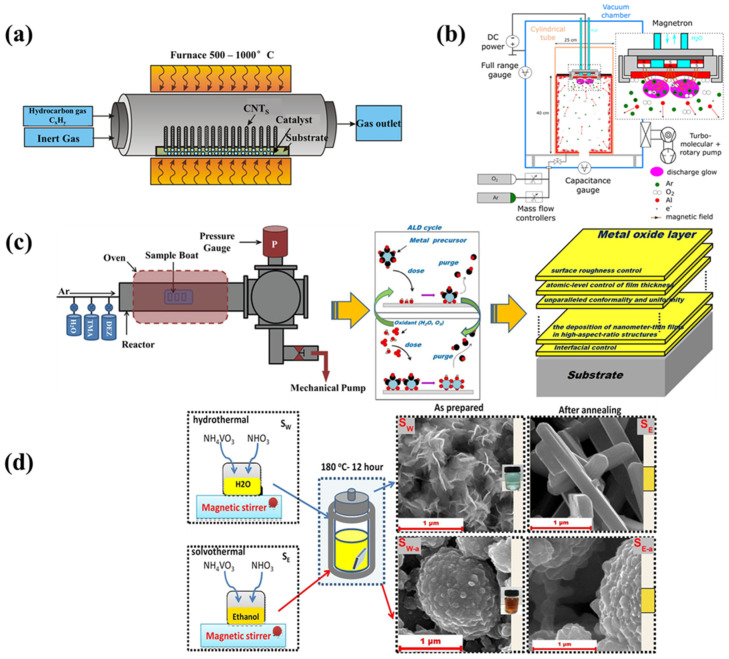
(**a**) Schematic diagram of chemical vapor deposition (**b**) magnetron sputtering (**c**) Atomic layer deposition and (**d**) hydrothermal and solvothermal [38,48,49,50,51,52].

## 4. Magnetic Storage Applications

### 4.1. Magnetic Storage Devices Using MgO in Heterostructures

Many academics have been interested in the Dzyaloshinskii Moriya Interaction (DMI) in recent years, because magnetism and chirality play a significant role in generating magnetic skyrmions and chiral domain barriers, which may be utilized to increase data storage [15,53]. Various MgO heterostructures their synthetic routes applications are shown in Table 1. The effective control of domain-wall motion can be achieved in advanced storage class memory devices by adjusting the thickness of the MgO. The magnetization and domain-wall motion in upright magnetized Ta/Pt/Co/MgO/Pt heterojunctions depend on the MgO’s thickness [54].

The obtained values, comprising two sets of specimens with opposing magnetic fields, show that as the MgO content increases, the equilibrium magnetic and repulsive fields decrease. However, adding an Mg layer minimizes saturation magnetization and coercive field. As shown in Figure 4b–e, the greatest diversity of loop sizes is seen in the loops with the thinnest magnesia and those without magnesia. Figure 4a shows cross-sectional images of the velocity field in a cylindrical domain with and without MgO. The velocity gradients are nearly quadratic, with the lowest gradients occurring at H_x_ non-zero. A comparable polynomial dependency of H_x_ on -ν can be seen when Hz is reversed, with the bare minimum occurring at the H_x_ inverse value. The black and red data points show that the MgO thickness exceeded a critical thickness, resulting in the DMI falling. The experimental findings of samples with Mg insertion layers indicated this the reduction in DMI is due to excessive oxidation. The diameter of the MgO layer grows as it becomes thicker. It is also possible that the DMI values for materials containing Mg will reveal that adjusting DMI results from an interfacial effect. Monoatomic Mg can be inserted into the MgO layer for tuning DMI and obtaining a high value of 2.32 mJ/m^2^ [55].

The L spin–orbit coupling that occurs in ultrathin magnetic heterostructures offers a new approach to electrically regulating magnetic moments. In a ferromagnet, the construction of a Neel-type domain wall may be driven by the spin Hall effect. Magnetic scale length and the evolution of magnetic storages devices are shown in (Figure 5).

It has been claimed that the Dzyaloshinskii–Moriya interaction plays a significant role in its formation. When the layer of metal next to a magnetic one is altered, the magnitude and indication of the domain wall motion (DMI) in that magnetic layer can be altered. When we tried using underlayers made of various materials (X = Hf, Ta, TaN and W), we discovered that the domain wall goes either with or against the transfer of electrons depending on the kind of component that was utilized. When the same underlayer material is used for the top and bottom layers, it was discovered that the sign of the persistence for the bulk spin–orbit coupling is still identical. However, when a different material is used, the sign of the bulk spin–orbit coupling constant changes. In (Figure 6a), a domain wall is shown as moving toward the positive X direction; this is true not just for Hf and Ta underlayer films, but also for thin TaN underlayer films. The domain wall in these films always travels in the same direction as the electron flow. For thicker TaN underlayer films and all W underlayer films, the movement of the domain wall is in the opposite direction of the flow of electrons [57]. The Dzyaloshinskii–Moriya interaction is seen here as a function of the magnitude of the voltage pulse in Figure 6b,c.

Due to the complexity of determining the structural symmetry of the interfaces, treating the origin of the DMI at the interface is more challenging. On the other hand, it was reported that DMI’s sign changes depending on how the films are stacked [76,77], which is under the use of the three-site indirect exchange [78]. Recent experiments [79] have shown that the DMI does not change even when a nearby non-magnetic layer (Pt or Ta) has the opposite sign of the spin–orbit coupling constant.

### 4.2. Magnetic Anisotropy of MgO Heterostructure

Strain significantly impacts the MCA value and causes the easy magnetic axis switch of the rotational coupling between states that are occupied with d_x2-y2_ and ones that are vacant with d_xy_. Investigating the energy and k-resolved dispersion of orbital features of the minority-spin band produced from Fe at the FeCo/MgO interface contributes considerably to PMA at zero strain. Research on the energy and k-resolved dispersion of the orbital character and strain-induced alterations of spin–orbit-related d-states may reveal the nature of the strain effect [62]. Magnetization shifting may be generated by spin-polarized current by the use of the spin-transfer torque (STT-RAM) [80] or by an electric field employing the magneto-electric effect (MeRAM) in magnetic tunnel junctions (MTJ) along with substantial perpendicular magneto-crystalline anisotropy (PMA) for use in applications involving high-density nonvolatile random access memory (RAM) [81,82,83]. An MTJ comprises thin ferromagnetic (FM) films sandwiched between a magnesium oxide barrier and heavy metal electrodes. The component layers are subjected to significant strain due to large lattice mismatches. The spin–orbit coupling (SOC) connects the spin degree of freedom to lattice distortion. As a result, the strain can significantly alter the system’s magneto-crystalline anisotropy (MCA) and other magnetic properties. The epitaxial strain strongly affects the magnetic correlation length [84] and PMA of magnetic oxides [85]. Recent tests have also shown that epitaxial strain in thin layers of magnetic oxides and semiconductors produced on diverse substrates may rotate the easy magnetic axis [86,87,88]. Strain and compositional deformation were researched on MCA in thin-film and bulk Fe/Co alloys [89,90]. MCA is significantly reduced when the FM layer is subjected to expansive strain. Furthermore, at a critical strain value, the easy magnetic axis switches from perpendicular to in-plane [62]. The use of a slab supercell approach along (001) having three monolayers (MLs) of bcc Ta, three MLs of B2-type Fe/Co, seven MLs of rock salt MgO and a 15 Å thick vacuum region separating the periodic slabs simulates the epitaxial growth of the Ta/Fe/Co/MgO junction, as shown in Figure 7a [62].

In magnetic tunnel junctions (MTJ), a rare-earth-free ferromagnetic nitride can be used to create a high-spin polarization and curie temperature [91,92,93,94,95]. It was discovered that an epitaxial Fe_4_N film on a MgO substrate exhibited magnetic anisotropy in the same plane as the film. Epitaxial growth is possible with the Fe_4_N (001) films formed on MgO (001) substrates. Fe_4_N has a lattice constant of 3.795 Å with a cubic anti-perovskite structure [92]. In the over-oxidized Fe_4_N/MgO interface, an additional O atom is placed in the interfacial Fe_4_N layer. In the under-oxidized Fe_4_N/MgO structure, oxygen vacancies appear in the Fe-N interfacial region due to its high regularity, which is similar to the Fe/MgO structural research (Figure 8a,b) [96,97,98]. Interstitial redox and applied electric modulation can alter the magnetism of Fe_4_N/MgO heterostructures by changing the magnetic anisotropy, which depends on charge-mediated Fe d-orbital rearrangement (Figure 8c).

As shown in Figure 8d, Fe_4_N, the magnetic anisotropy of Fe I and Fe II atoms, has different response characteristics with the Fe II atom’s magnetic anisotropy being more easily affected by environmental factors. An electric field can induce magnetic anisotropy, making it easier to develop energy-efficient information storage. The Fe_4_N I-layer and V-layers and the Fe I and Fe II atoms have opposite MAE, whereas the Fe_4_N III-layer has the same sign of MAE, because of which the MAE of the interfacial Fe_4_N layer is so tiny. The magnetic moment of Fe I and Fe II atoms changes when the N atom influences the indirect exchange interaction between Fe atoms [61].

### 4.3. Perpendicular Magnetic Anisotropy

The current spin injectors (GaMnN) [99,100], GaCrN [101], Fe_3_O_4_ [102,103], and CoFe [104,105] without in-plane magnetization anisotropy cannot be used in area dispersion topologies such as 3D displays for deployments of spin-LEDs and spin lasers as shown in Figure 9a. This is because, in addition to creating light that is spirally oriented from the quantum-well (QW) LED surface, the spin injector’s magnetization must be kept perpendicular to the surface regarding optical directions [106]. If the spin injector’s magnetization is in-plane, we will need an external magnetic field of several Tesla to rotate it perpendicular to the formed layers. From tests and first-principles calculations, Gao and co-workers have shown that Au/Co/MgO/GaN heterostructures exhibit a high perpendicular magnetic anisotropy. The Au/Co/MgO heterostructures were epitaxially produced on GaN/sapphire substrates using molecular beam epitaxy (MBE). The presence of a substantial perpendicular magnetic anisotropy in a 4.6 nm thick Co on MgO/GaN is discovered. After examining two control samples, it was discovered that most of the PMA in the Au/Co/MgO/GaN heterostructure is caused by the Co/MgO interface. According to first-principles calculations on the Co (4 ML)/MgO structure, the MgO (111) surface can boost the PMA value by roughly 40% when compared to the pure 4 ML thick Co hcp lm [107]. Figure 9c shows the p-orbitals and d-orbital-resolved Ki values for the Fe/W/MgO, Fe/Re/MgO, Fe/Pt/MgO and Fe/Bi/MgO systems. One interfacial Fe atom on Fe/MgO on each side contact was substituted through X resulting in the Fe/X/MgO model. As a result, our model includes two symmetric X/MgO contacts [108] (Figure 9b). To check the quality of the layer’s in situ reflections, high energy electron diffraction (RHEED) measurements were performed with the incoming electron beam accelerated at 30 kV. In addition, Safdar et al.’s first-principles calculations were used to investigate interfacial perpendicular magnetic anisotropy and cleavage in Fe/X/MgO structures, wherein X corresponds to one layer of lanthanides. Her findings indicate that W, Re and Pt are promising heavy elements for improving perpendicular magnetic anisotropy in Fe/MgO. The enhanced KI values for these elements are 2.43 mJ/m^2^, 2.37 mJ/m^2^ and 9.74 mJ/m^2^, respectively. This work shows that by adding heavy components at the Fe/MgO interface, interfacial engineering is viable for improving interfacial perpendicular magnetic anisotropy [108].

### 4.4. Anisotropy Voltage Control

In magnetic heterojunctions, regulating magnetic anisotropy by applying voltage (VCMA) is critical for creating energy-efficient digital equipment with absurdly low energy usage, such as MRAMs [109,110]. Magneto-tunnel junctions (MTJs) are essential components of high-performance spintronic devices [111]. Voltage-controlled magnetization switching in MTJs has tremendously benefited the VCMA technique [66,67,83,112,113,114,115,116,117,118,119,120]. The low power consumption of state-of-the-art CoFeB/MgO MTJs is achieved by controlling their magnetization with a low-voltage magnetic field [66,67]. The CFA/MgO connection induces a substantial perpendicular magnetic anisotropy (PMA) in the Ru/Co_2_FeAl (CFA)/MgO magnetic heterojunctions [121,122,123]. First-principles calculations suggest that the CFA/MgO structure could exhibit a very high VCMA effect of up to 1000 fJ/Vm [124]. Boron dispersion in Ta/CoFeB/MgO compounds may result in unanticipated reactions and/or inter-layer diffusion [125,126]. This is significant for both the electrode surface PMA and VCMA effects. The interfacial atomic structures of growing extremely thin CFA/MgO (001) heterostructures have recently been studied. Tunnel couplings with the Ru/CFA/MgO structure were created and magnetized perpendicularly. Such junctions in the CFA layer demonstrated good voltage control on switching fields. The CFA film has a VCMA coefficient of 108 fJ/Vm at ambient temperature and 4 K [65].

Figure 10a depicts the entire stack structure of intended p-MTJs. In addition to each layer’s deposition state, the voltage orientation is also recorded. When AC voltage is applied between the top and bottom layers, the excellent orientation of the voltage source is determined. The reference layer’s usage of CoFeB/Ta bilayers is intended to boost the MR ratio by enhancing comprehensible tunneling over the MgO block: tunneling resistance (R) vs. the out-of-plane magnetic field (H) for a designed p-MTJ with a 100 nm dimension. A direct-current (dc) voltage of 1 mV was employed to assess the resistance. The resistance-area product of the sample was 175 Ω µm^2^. A TMR ratio of 65% was attained. At TMR, significant cycles were adjusted relative to the magnetic field at voltage levels of 800 mV, 1 mV and 800 mV. The structure in Figure 10b depicts an orthogonally magnetized MTJ with the Ru/CFA/MgO/CoFeB structure. The normalized TMR curves are plotted against the applied magnetic field under an in-plane magnetic field at various voltages (the positive field area is displayed in this figure). The CFA magnetizations are a typical normalized in-plane component. The darkened region may be used to determine the PMA energy density. The magnetic anisotropy K_u_t for the CFA film relates to the electric field (Figure 10c). The MTJs were alternatively exposed to 1 V bias voltages 300 times. The drawings show the magnetic arrangement when a voltage is provided to the MTJ. The inset shows the magnification used for 50 measurements. When a voltage is applied, a significant change occurs in the CFA layer’s switching field (Hs). The voltage dependence of the Hs relaxation rate for the CFA layer from the P-phase to the AP-phase magnetization states changes. Increasing the voltage from the negative to positive bias direction decreases coercivity [65].

### 4.5. Spin–Orbit Effects in MgO Heterostructures

Compared to conventional spin–transfer torque, current-induced spin-orbit torques (SOT) have also been used recently in order to alter the magnetization of an extremely thin ferromagnetic (FM) in a way that is effective in terms of energy use [63]. The current is routed through a double layer comprising an FM and a substance by strong spin–orbit coupling (SOC) such as a heavy metal (HM) [127,128,129] and a topological insulator in a SOT device [130,131]. Different factors, including the edge value spin Hall direction of the origin of the SOT signal [129] and the type of oxide capping the bilayer [132,133], determine the amount of SOT for a given system. For the following reasons, Pt/CoFeB is a viable contender for implementations using optoelectronic systems. It can be quickly grown on substrates such as Si or AlO_x_, and its magnetic properties can be tuned by alloying the CoFeB layer [134]. Pt is a highly conductive material that reduces joule heating and power consumption [135]. Due to its substantial spin divergence in combination with utilizing crystal-like MgO, a CoFeB compound is a broadly used FM in spintronic expedients [136]. CoFeB/MgO-based tunnel junctions have achieved tunnel-based sensor values of up to 600 percent at room temperature [137] (Figure 11a). Applying Pt (5 nm)/Ti(t_Ti_)/CoFeB(1 nm)/MgO Hall bar nanostructures, the influence of interfacial change following Ti inclusion on SOT-induced magnetic permeability transitions was examined. We may draw two conclusions from the SOT-induced switching data. One is that all samples, regardless of t_Ti_, have the same switching polarity. Under a positive magnetic field, a positive (negative) current promotes magnetization in the downward (upward) direction. The crucial switching current (1/I_C_) and the SOT switching quality, which is expressed by the crucial applied voltage normalized by the anisotropy field (J_C_/B_k_)^−1^, is a temperature-dependent function of time t_Ti_, as shown in Figure 11(ai,aii). When the magnetic permeability is in resonance, a spin current is injected from the CoFeB coating into the Pt layer, resulting in a rise in the effective damping constant effect in the CoFeB layer and a transverse electric voltage resulting from the inverse spin Hall effect (V_ISHE_) in the Pt layer. The spin current causes the former to be injected into the Pt layer, whereas the latter is caused by the total spin current lost in the CoFeB layer. The normalized V_ISHE_ Pt thickness is dependent on the sample resistance and Δαeff respectively, as shown schematically in Figure 10b. When an AC is applied to a sample, the Δ_BDL_ and Δ_BFL_ are measured, and oscillations in magnetization are observed. The second-harmonic Hall voltages for Pt/Ti(t_Ti_)/CoFeB/MgO testers by several t_Ti_ values are shown in V_x_^2ω^ and V_y_^2ω^. The plots above show the results of the Pt/CoFeB/MgO sample V_x_^2ω^ (solid symbols) and V_y_^2ω^ (open symbols), namely, XMCD, integrated spectra and current-induced switching curves. A Pt/CoFeB/MgO structure has been used to make domain barriers, DC rotational devices, and skyrmion vibrational strategies, as shown in Figure 11(bi–biii) [56,138,139,140].

## 5. Energy Storage Applications

### 5.1. Energy Storage Devices

The excessive usage of fossil fuels necessitates the development and emergence of renewable fuels into everyday life. Battery technology has improved over the past two decades, allowing for the development of batteries (e.g., lithium-ion batteries, sodium-ion batteries, and potassium-ion batteries (KIBs)), super capacitors, and lithium sulfur batteries to store and use electricity at any time. The electrode materials significantly influence the energy storage capability of these devices. Graphite, hard carbon, metal oxides, sulfides and carbides have all been investigated as potential energy storage materials with intriguing electrochemical performance when used as an electrode [1]. The heterostructure is formulated on the basis of a combination of various semiconductor materials with comparable crystal shapes, atomic distances and thermal expansion coefficients. The differing band structures, variations in predominant conductivity and Fermi level differences can cause heterostructures to develop interfaces between different layers. This is because the chemical composition and charge distribution vary across these interfaces [5].

### 5.2. Heterostructures’ Energy Storage Mechanisms

The benefit of heterostructure is more than merely integrating different materials with varying strengths. In general, the microstructure of heterointerfaces is strongly related to the distinctive advantages of the heterostructure. When two building blocks come into contact, band alignments occur spontaneously at heterointerfaces, causing charge redistribution around the interfaces (Figure 12) unless the Fermi energies of certain elements become balanced. As a result, the holes plus electrons are divided via an ultimately charged area due to the built-in potential around the heterointerface. Because of this built-in potential, the heterostructures stimulate forward bias and practically insulate against reverse discrimination. The predominant differences in carrier density and the resultant electric field also impact charge redistribution behavior (BIEF). Carriers should go from the building block with the highest majority carrier concentration to the one with the lowest. The formed BIEF will act as an anti-force, preventing transport spread; as a result, band topologies, carrier densities and BIEF intensity influence the final state of charge redistribution. During the alignment process, both bands with adjacent essential components would be curved with a particular offset, and the entire bending would then be determined as the ability. The distribution of the charge can be controlled by the configuration of energy bands surrounding heterointerfaces, which are categorized as a straddling gap (type-I), staggered gap (type-II) and broken gap (type-III). In the type-I alignment, photon-generated carriers gather in a similar building block, decreasing its bandgap [2].

Wang and coworkers found that they could develop constant VTe_2_ shells over MgO particulate centers by adapting CVD synthetic conditions, resulting in a VTe_2_@ MgO heterostructure that could be used to regulate LiPS. The metal VTe_2_@MgO created using the in situ vapor-phase processes is advantageous for synergizing the double functions of VTe_2_ and MgO because it has a clean and unbroken interface. The resulting S/VTe_2_@MgO positive electrode shows long-standing cycling capability with a performance degradation of just 0.055 percent each cycle across a thousand processes at 1.0 C. Even containing a sulfur dose of 6.0 mg/cm^2^, the positive electrode still offers attractive electrochemical performance that matches the best high-loading LiS systems.

The researchers developed a straightforward and scalable CVD technique for fabricating VTe_2_@MgO heterostructures that function as sulfur-hosting promoters within the LiS battery domain. At various annealing temperatures, metallic VTe_2_ shells may envelop MgO cores, producing VTe_2_@MgO heterostructures that allow effective LiPS control to increase the sulfur reaction kinetics. The CVD synthesis permits the ascendable creation of VTe_2_ materials, in which a white MgO powder turns black when grown, revealing uniform VTe_2_ growth on the surface of MgO. Four-probe measurements were performed to assess the conductivity of VTe_2_@MgO prepared at various heat levels to reveal the metallic nature of as-grown VTe_2_. The heterostructure produced at 650 °C has the greatest electrical conductivity (2.6 S m^−1^ 1).

SEM, TEM, Raman spectroscopy and XPS confirm the successful VTe_2_ synthesis on MgO (Figure 13a–n). Sulfur redox kinetics were analyzed by CV profiles of sulfur cathodes in various electrolytes. The asymmetric cell containing CP-VTe_2_@MgO was also prepared and tested in 0.2 M Li_2_S_6_ electrolyte at 50 mV/s. Identical cell patterns with or without 0.2 M Li_2_S_6_ electrolyte at a 0.5 mV/s scan rate were recorded for comparison. Plots were made showing the CV peak current vs. scan rate square root and the Li^+^ diffusing energy profile on the VTe_2_ (011) facet. A diagram of the polysulfide regulating process was created using a VTe_2_@MgO heterostructured promotor. Galvanostatic charge/discharge patterns at speeds ranging from 0.2 to 3.0 C. S/VTe_2_@MgO cathode cycling performance were observed at 1.0 C. A comparison was made of the areal capacities of high-loading cathodes between our study and other sulfur hosts previously described (Figure 14a–g). As a result, S/VTe_2_@MgO cathodes exhibit favorable electrochemistry with a low-capacity decline of 0.055% per cycle over 1000 cycles at 1.0 C. Furthermore, a sulfur loading of 6.0 mg cm^−2^ was found to yield a high-loading S/VTe_2_@MgO cathode with a theoretical areal capacity of 5.9 mAh cm^−2^ in their study. Their findings reveal the spotless and precise preparation of heterojunction mediators with unusual electrocatalytic activity, opening the way for developing multifunctional materials [71].

## 6. Conclusions and Prospects

Various heterostructures exhibit beneficial relationships between two or more building elements, making them work better. To achieve multifunctionality, each component plays a role that helps the whole. MgO is a promising candidate for multifunctional devices, as it has unique properties that enable it to perform new functions. Furthermore, MgO’s unique electronic and magnetic properties result from complicated interactions among its electron orbitals, crystal lattice vibrations and electron spins. When MgO is combined with ferroelectric materials to form multiferroic heterostructures, magneto/electrostrictive interactions, exchange interactions and spin–orbital coupling emerge. These properties result in novel physical properties. Interfacial engineering becomes crucial in MgO-based heterostructures, including charge transfer, magnetic anisotropy, oxygen vacancy and strain effects. These novel physical properties could be helpful in nan-electronic, spintronic, optoelectronic and energy storage devices.

Nonetheless, understanding the nature of heterostructures is still in its early stages. One question scientists are still debating is whether heterostructures can be used to store more energy. Future research should focus on determining which mechanism dominates the energy storage process, how charge redistribution and BIEF affect the overall energy storage performance, and what is the best setting for achieving optimal heterostructure energy storage results. Additional solid mechanism research and experimental investigations should be conducted for a more accurate understanding. Machine learning is being used to screen materials and predict their performance, and COMSOL Multiphysics modeling can help to understand the electromagnetic charge distribution in electrodes.

Meanwhile, newer material characterization techniques such as in situ characterization techniques, nuclear magnetic resonance and cryogenic electron microscopy have been investigated and employed in research on energy storage mechanisms. These new theoretical calculation approaches and characterization methodologies might be valuable tools for researchers working on improved heterostructure electrode mechanisms. Besides their fundamental nature and properties, the impact of structural factors on electron/ion transfer kinetics and energy storage performance (such as heterointerface locations and sizes, chemical bonds or van der Waal’s force coupling forces, and transition layer thickness between building blocks) is underdeveloped. Researchers building high-performance heterostructures can benefit from a better knowledge of these structural elements and their underlying nature.

## Figures and Tables

**Figure 1 nanomaterials-12-02668-f001:**
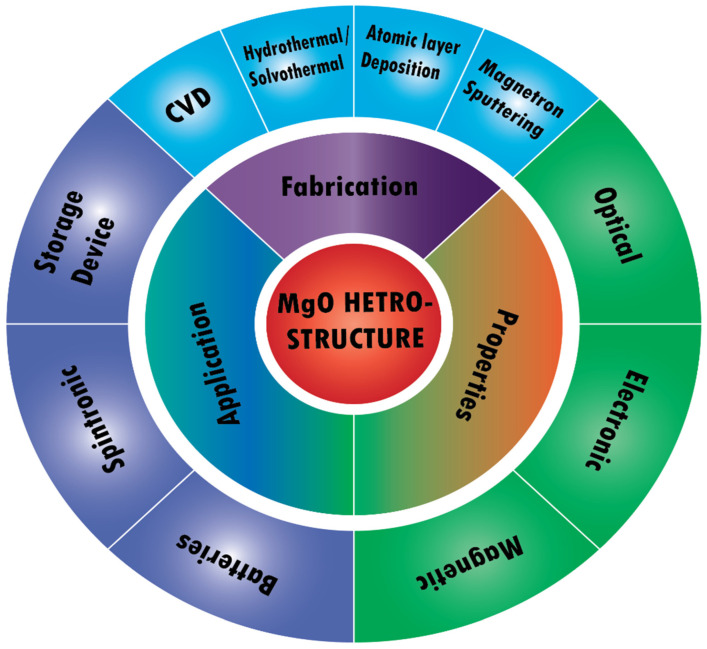
Schematic illustration of MgO heterostructure applications, properties and fabrication.

**Figure 4 nanomaterials-12-02668-f004:**
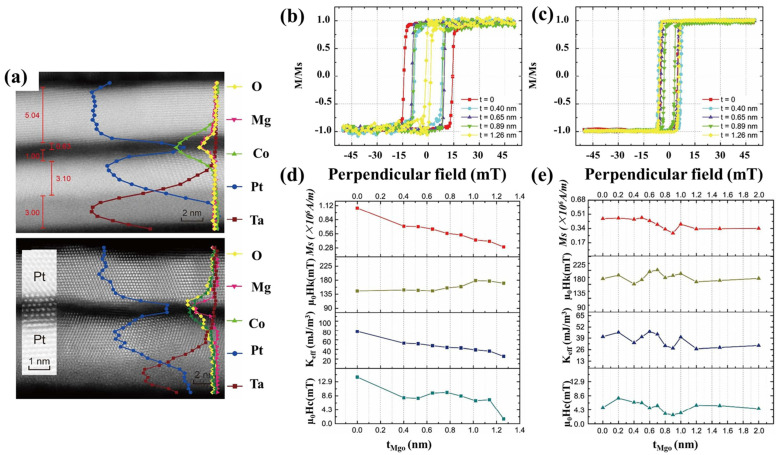
(**a**) Measurements of the cross profile of the samples without Mg layer at t 0.60 nm and with Mg layer at t 0.80 nm using transmission electron microscopy. (**b**) the Ta/Pt/Co/MgO(t)/Pt stacks and (**c**) the Ta/Pt/Co/Mg(0.2 nm)/MgO(t)/Pt stacks with various MgO thicknesses, hysteresis loops with perpendicular magnetic fields were observed. Hysteresis loops for samples (**d**) without and (**e**) with Mg insertion layer yielded magnetic characteristics [55].

**Figure 5 nanomaterials-12-02668-f005:**
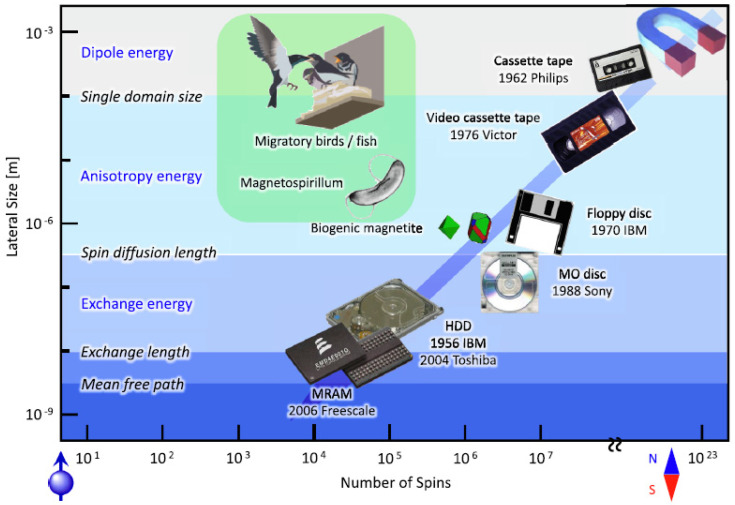
Magnetic length scales and the evolution of magnetic storage technologies [75].

**Figure 6 nanomaterials-12-02668-f006:**
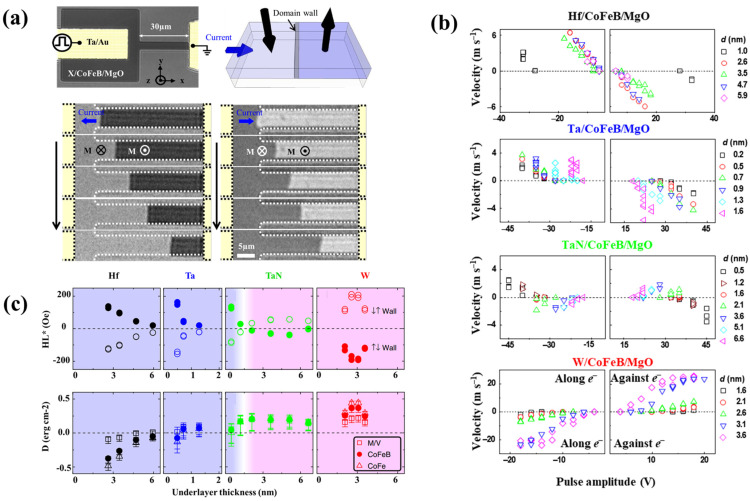
(**a**) Magneto-optical Kerr pictures exhibiting current-induced domain wall motion and a schematic of the experimental apparatus. (**b**) For magnetic heterostructures with four different underlayers, domain wall velocity as a function of pulse amplitude is displayed. Hf, Ta, TaN (Q: 0.7%) and W make up the underlayer. (**c**) Underlayer-dependent Dzyaloshinskii–Moriya interaction [57].

**Figure 7 nanomaterials-12-02668-f007:**
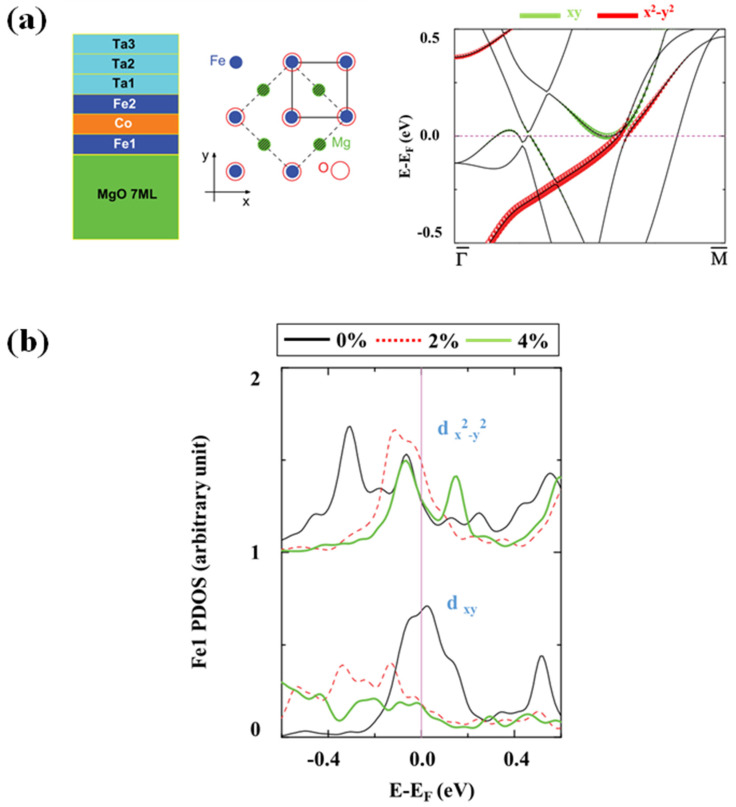
(**a**) Ta/Fe/Co/MgO heterostructure schematic model. Atomic arrangement at the Fe/Co/MgO contact with O atoms stacked on top of Fe atoms. For Fe1 d_xy_ (green) and d_x_^2^_y_^2^ (red), energy and k-resolved distribution of orbital characteristics of minority spin bands (red). (**b**) Density of states (PDOS) of minority Fe1 d_xy_ and d_x_^2^_y_^2^ under 0%, 2% and 4% strain.

**Figure 8 nanomaterials-12-02668-f008:**
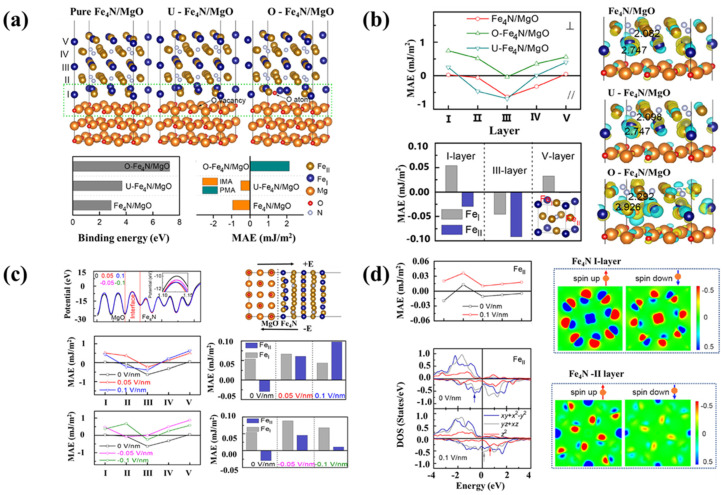
(**a**) A series of Fe_4_N/MgO heterostructures were fabricated with varied interfacial conditions. I–V are the layer numbers of Fe_4_N. Binding energies and magnetic anisotropy energy of Fe_4_N/MgO heterostructures with varying interfacial conditions. (**b**) Layer-resolved magneto-acoustic emission (MAE) of Fe_4_N/MgO heterostructures with varying interfacial conditions. (**c**) The MAE of Fe (I) and Fe (II) atoms in Fe_4_N I-, III-, and V-layers of the pure Fe_4_N/MgO heterostructure is presented. In the inset, we can see the structure of bulk Fe_4_N. (**d**) Charge density differences in Fe_4_N/MgO heterostructures with varying interfacial conditions can be visualized by computing the isosurface value of 0.15 e/ Å^3^. The magnetic moments of certain Fe atoms are highlighted in the figure [61].

**Figure 9 nanomaterials-12-02668-f009:**
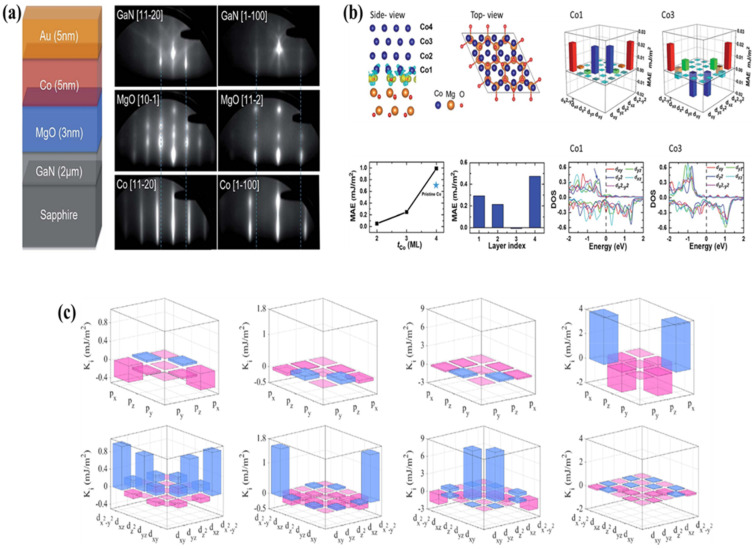
(**a**) A diagrammatic representation of the heterostructure consisting of Au, Co, MgO, and GaN and in situ RHEED designs. It has been discovered that the epitaxial connection between Co (1120)/MgO (101)/GaN (1120) and Co (1100)/MgO (112)/GaN (1100) is epitaxially connected. (**b**) The example of the many design techniques. Co1 and Co3 orbitals resolved MAE and the impact of their doping on the density of states in Co_4_ML/MgO [108] (**c**) Calculated orbitals–resolved Ki for interfacial atoms [107].

**Figure 10 nanomaterials-12-02668-f010:**
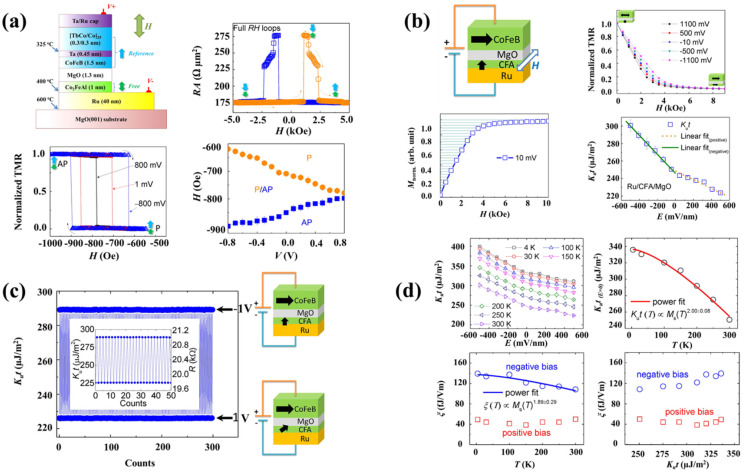
(**a**) The VCMA impact in p-MTJs is demonstrated. (**b**) Quantitative assessment of the VCMA impact in Ru/CFA/MgO heterostructures. (**c**) Measuring the VCMA impact again and again. (**d**) Temperature and magnetic anisotropy effects on the VCMA effect in Ru/CFA/MgO heterostructures [65].

**Figure 11 nanomaterials-12-02668-f011:**
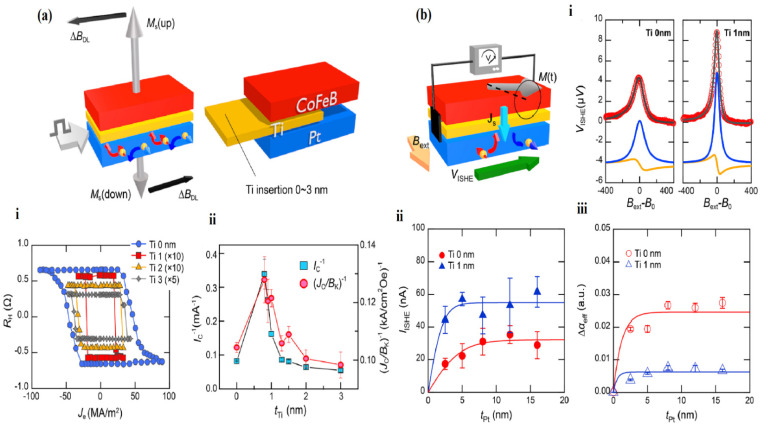
(**a**) Schematic diagram of the spin–orbit torque (SOP). (**i**) The current-induced switching curves with reference to Ti thickness. (**ii**) The Pt/Ti(t_Ti_)/CoFeB/MgO (3.2 nm) structures’ switching performance expressed as the reciprocal of I_C_ (**left**) and J_C_/B_K_ (**right**). (**b**) Schematic of the measurements: (**i**) V_ISHE_ values in micrometers. (**ii**) For Pt (t_Pt_)/Ti(0, 1 nm)/CoFeB/MgO samples, V_ISHE_ was normalized by sample resistance (I_ISHE_) vs. Pt thickness t_Pt_. (**iii**) Damping constant [56].

**Figure 12 nanomaterials-12-02668-f012:**
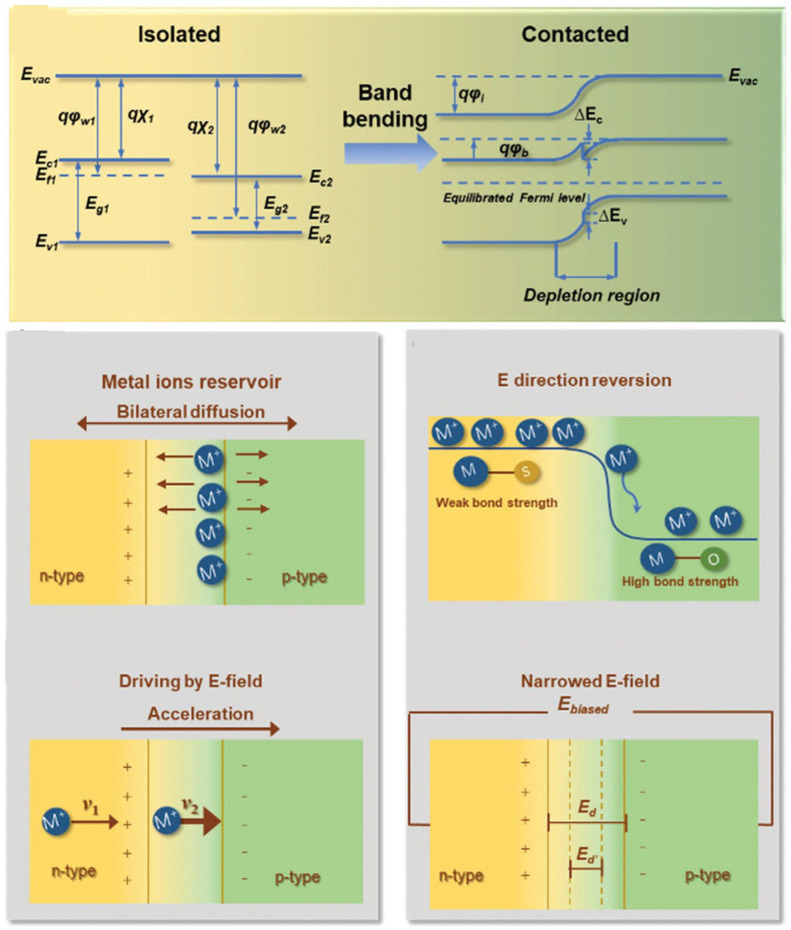
Heterostructure bond alignment and energy storage methods diagram [2].

**Figure 13 nanomaterials-12-02668-f013:**
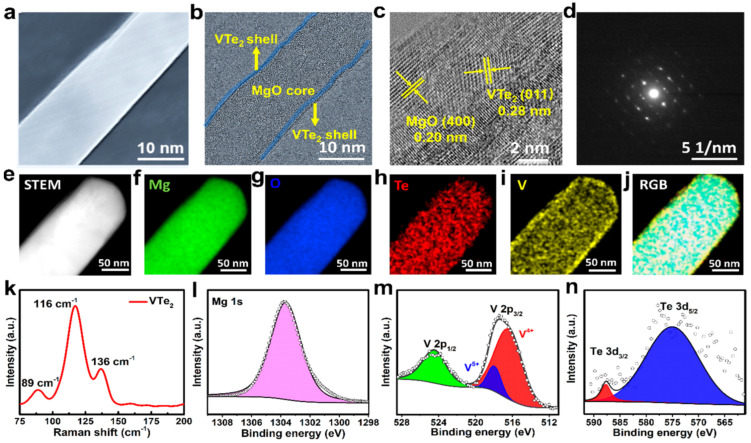
Structural and elemental analysis. (**a**) SEM. (**b**–**d**) TEM. (**e**–**j**) STEM. (**k**) Raman spectra. (**l**–**n**) XPS spectrum [71].

**Figure 14 nanomaterials-12-02668-f014:**
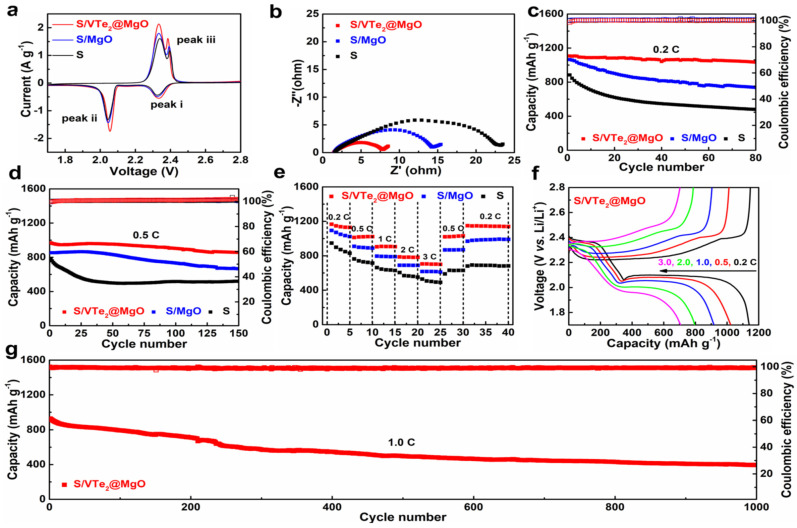
Electrochemical performances of S/VTe_2_@MgO. (**a**) CV profiles. (**b**) EIS curves. (**c**,**d**) Cyclic performance at 0.2 and 0.5 C. (**e**) Rate performances of S/VTe_2_@MgO. (**f**) Galvano static charge–discharge profiles. (**g**) Cyclic performance at 1.0 C [71].

**Table 1 nanomaterials-12-02668-t001:** Various heterostructures of MgO: synthetic strategies and applications.

Sr No	Material	Developing Method	Application	Reference
1	Ta/Pt/Co/MgO/Pt	magnetron sputtering	tuning of DMI	[55]
2	Pt/CoFeB/MgO	magnetron sputtering	magnetization switching	[56]
3	X/CoFeB/MgO	magnetron sputtering		[57]
4	Au Embedded ZnO/heterostructure	hydrothermal and citrate reduction methods	photocatalytic activity	[58]
5	Au modified SrTiO_3_/TiO_2_	hydrothermal post-photoreduction method	photocatalytic activity	[59]
6	steel mesh embedded MgO nanowires	conventional evaporation	high-emission current density	[60]
7	Fe_4_N/MgO	exchange correlation potential	switching magnetic anisotropy	[61]
8	Ta/FeCo/MgO	first-principles density functional	strain control on magnetocrystalline anisotropy	[62]
9	Hf/CoFeB/MgO	magnetron sputtering	spin-orbit torques	[63]
10	Au/FeCo/MgO	magnetron sputtering	VCMA behavior	[64]
11	Ru/Co_2_FeA/MgO	deposition method	voltage control on switching fields	[65]
12	CoFeB/MgO	magnetron sputtering	reduce the switching energy	[66]
13	CoFeB/MgO	physical vapor deposition	electric field-induced switching with energies	[67]
14	Cr/Fe/MgO	electron beam evaporation	coefficient of the electric field effect	[68]
15	MgO/NCS-CC between MgO and NiCo_2_S_4_	electrodeposition hydrothermal and annealing	oxygen evolution reaction activity	[69]
16	Ta/CoFeB/Mgo	annealing	negative influence on the crystallization of CoFe	[70]
17	VTe_2_@MgO	CVD and vapor-phase	electrocatalytic activity for LiPS regulation	[71]
18	tungsten doped single crystal Fe/MgO	radio frequency and direct current sputtering	PMA and voltage effect	[72]
19	Fe/MgAl_2_O_4_(001)	electron beam evaporation	perpendicular magnetic tunnel junctions and theoretical predictions.	[73]
20	β-Ga_2_O_3_/MgO	thermal evaporation	phototransistor with ultrahigh sensitivity	[74]

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
