# Peer review of "MgO Heterostructures: From Synthesis to Applications"

_nanomaterials, 2022, doi:10.3390/nano12152668_

Round 1

Reviewer 1 Report

referee report 

nanomaterials-1771134-peer-review-v1 

Role of MgO in Heterostructures: Synthetic strategies and Applications in Magnetic Storage Devices and 

Energy Storage

Tabasum Huma et al.

The present manuscript has received a new number from MDPI, so I would expect to see some substantial changes

according to the previous comments. However, I do not see an author's reply nor really an improvement of the

manuscript. Yes, we do now have only 140 references instead of 159 before (why that?).

So, I can repeat here my previous comments: 

In the present manuscript is a review discussing the role of MgO in heterostructures. Given the huge importance

of this material for magnetic storage devices and energy storage, this is an interesting topic being well suited

for publication in Nanomaterials.

The present manuscript comprises 13 figures with many sub-figures, 1 table and 159 references are given, which 

provide a good overview on the field.

However, the manuscript is by no means well prepared or arranged, which requires a lot of additional work. Many

of these problems are so obvious, so the authors should have noticed prior to submission. All this makes reading

of the manuscript in its present form quite tedious.

Here, I list several of these points:

# Please format ALL chemical formulae in the text properly, but ALSO in the article titles in the reference list.

# Figure 1 does not have a number nor a caption.

# Please take care for spaces in the entire manuscript. Very often, the text collides with references/fullstops, etc.

# Often, references are separated from the sentence by ".", which does not correspond to the journal style.

# Please take care that abbreviations are properly defined. When defining, the full name should be given followed by

the abbreviation in (). Why CVD in Sec. 4 has a "." in front?

# It is uncommon to use wordings like "you" in a scientific text.

# The English requires substantial improvement throughout the manuscript.

# All figures with their many subfigures must be presented with much higher resolution. Then one may check if they

are suitable for publication.

# References within the main body of the manuscript do not require the year of publication.

# Physical units do have proper abbreviations, so also use them and do not spell them out.

# The reference list does not follow the journal style.

# The references are not given in a consequent manner -- the style varies throughout the list. Please unify this.

# Very important for a review article are proper figure captions. It is often not sufficient just to copy the caption

text from the original publication as it is often necessary to provide a deeper information. So, take care to properly

describe EVERYTHING which is shown in the graph.

---> Especially the last point should be carefully considered by the authors. Yes, I can understand why one wants to

    have a figure with many subfigures, but the caption to it MUST BE WRITTEN PROPERLY so that all information visible

    is properly described.

Thus, considering this, and the fact that the figures still do not have a proper resolution, a MAJOR REVISION is still necessary.

Reviewer 2 Report

Reviewer’s response

Comments to the Author

In this manuscript, the authors reviewed the studies on the role of MgO as magnetic and energy storage materials. MgO is a well-known insulator with a wide bandgap of approximately 8 eV, which can be used in insulation applications with its low heat capacity and high m.p.. Besides, anti-sputtering properties, high transmittance, and secondary electron emission coefficient indicate that MgO is able to be used as protective layers for dielectric materials. Further, the authors introduced fabrication methods and previous studies of MgO for applications. In my opinion, if some minor revisions should be properly addressed, this review will be a good guidance for the research and development of the applications of MgO and be accepted for publication in this journal.

1. This review seems to be mainly focused on the magnetic applications of MgO. Therefore, the title should be revised to emphasize this description.

2. To smoothly develop the author’s explanation, I suggest merging phrases 3.5, 3.6, and the introduction of magnesium oxide [see page 3, line 20].

3. I suggest revising the phrase as the following description.

1. Introduction

2. Metal oxide

3. Magnesium oxide

3.1 Basic

3.2 Properties

3.3 Fabrication techniques

4. Magnetic storage applications

  4.1 Magnetic Storage Devices using MgO in Heterostructures

  4.2 Magnetic Anisotropy of MgO heterostructure

  4.3 Perpendicular Magnetic Anisotropy

  4.4 Anisotropy voltage control

  4.5 Spin Orbit Effects in MgO Heterostructures

5. Energy storage applications

  5.1 Energy storage devices

  5.2 Heterostructures' Energy Storage Mechanisms

6. Conclusion and Prospects

4. Most figures have fuzzy texts. Thus, the figures should be revised with high resolution.

5. Several figure captions should be minorly revised.

1) In caption for Figure 1, please revise from Figure 1: to Figure 1.

2) In caption for Figure 2, please do spacing between “diagram” and [14].

3) In caption for Figure 3, please merge the reference form as [25, 26-29].

4) In caption for Figure 4 and 6, please remove one of period [see caption for Figure 4 “~ in a perpendicular magnetic field .[54].; Figure 5 “~ interaction. [56].]

5) In caption for Figure 8, please do spacing between “figure” and [60].

6) In caption for Figure 9, please revise capital and subscript.

7) In caption for Figure 10, please do spacing between “heterostructures” and [126].

8) In caption for Figure 12, please do spacing between “diagram” and [2].

9) In caption for Figure 13, please revise the first character to capital.

10) In caption for Figure 14, please remove or revise the over-spacing between “cyclic” and “performance”.

6. In the manuscript, please recheck the full name of terms and abbreviations.

7. In the manuscript, please recheck the following article style (e. g. Figure index in the phrase; Fig. 1, Fig. 2, or Figure 1, Figure 2).

8. It seems that the English of the full text slightly needs further polishing. Please proofread this manuscript again before submission.

Author Response

This manuscript is a resubmission of an earlier submission. The following is a list of the peer review reports and author responses from that submission.

Round 1

Reviewer 1 Report

This is a review paper about the role of MgO in heterostructures. The review is badly written, with several English errors and no attention payed to make the text clean and accessible. No care is taken in writing the review, as it is apparent for the number of typos I found in the draft. The structure of the review is not well organized, with some sections focused on MgO, some other sections very general, and apparently useless to develop the main theme.

For example, it is not clear why, in the introduction, the authors talk exclusively about wires, which is only one of the possible nano-objects and not relevant for the review. As another example, in the second section of their manuscript, the authors say that an heterostructure is made up of a mixture of two different semiconducting materials. This is false. It can be made also by more than two different semiconductors.

Almost all figures have a very poor resolution, and they are excessively crowded.

I do not recommend the publication of this manuscript.

Reviewer 2 Report

referee report 

nanomaterials-1771134-peer-review-v1 

Role of MgO in Heterostructures: Synthetic strategies and Applications in Magnetic Storage Devices and 

Energy Storage

Tabasum Huma et al.

In the present manuscript is a review discussing the role of MgO in heterostructures. Given the huge importance

of this material for magnetic storage devices and energy storage, this is an interesting topic being well suited

for publication in Nanomaterials.

The present manuscript comprises 13 figures with many sub-figures, 1 table and 159 references are given, which 

provide a good overview on the field.

However, the manuscript is by no means well prepared or arranged, which requires a lot of additional work. Many

of these problems are so obvious, so the authors should have noticed prior to submission. All this makes reading

of the manuscript in its present form quite tedious.

Here, I list several of these points:

# Please format ALL chemical formulae in the text properly, but ALSO in the article titles in the reference list.

# Figure 1 does not have a number nor a caption.

# Please take care for spaces in the entire manuscript. Very often, the text collides with references/fullstops, etc.

# Often, references are separated from the sentence by ".", which does not correspond to the journal style.

# Please take care that abbreviations are properly defined. When defining, the full name should be given followed by

the abbreviation in (). Why CVD in Sec. 4 has a "." in front?

# It is uncommon to use wordings like "you" in a scientific text.

# The English requires substantial improvement throughout the manuscript.

# All figures with their many subfigures must be presented with much higher resolution. Then one may check if they

are suitable for publication.

# References within the main body of the manuscript do not require the year of publication.

# Physical units do have proper abbreviations, so also use them and do not spell them out.

# The reference list does not follow the journal style.

# The references are not given in a consequent manner -- the style varies throughout the list. Please unify this.

# Very important for a review artilce are proper figure captions. It is often not sufficient just to copy the caption

text from the original publication as it is often necessary to provide a deeper information. So, take care to properly

describe EVERYTHING which is shown in the graph.

To summarize, the present manuscript shows very interesting material, but is far from being well prepared. The present

manuscript is by no means suitable for publication in its present form. As the material and the topic is important, 

the manuscript should be reconsidered after a MAJOR revision.